# Mechanism of action for small-molecule inhibitors of triacylglycerol synthesis

Xuewu Sui [1,2,3], Kun Wang[1,2], Kangkang Song[4,5], Chen Xu[4,5], Jiunn Song[1,2], Chia-Wei Lee[1,2], Maofu Liao [2,6,10] ✉, Robert V. Farese Jr. [1,2,7,8,10] ✉ & Tobias C. Walther [1,2,7,8,9,10] ✉

Inhibitors of triacylglycerol (TG) synthesis have been developed to treat metabolism-related diseases, but we know little about their mechanisms of action. Here, we report cryo-EM structures of the TG-synthesis enzyme acyl-CoA:diacylglycerol acyltransferase 1 (DGAT1), a membrane bound O-acyltransferase (MBOAT), in complex with two different inhibitors, T863 and DGAT1IN1. Each inhibitor binds DGAT1's fatty acyl-CoA substrate binding tunnel that opens to the cytoplasmic side of the ER. T863 blocks access to the tunnel entrance, whereas DGAT1IN1 extends further into the enzyme, with an amide group interacting with more deeply buried catalytic residues. A survey of DGAT1 inhibitors revealed that this amide group may serve as a common pharmacophore for inhibition of MBOATs. The inhibitors were minimally active against the related MBOAT acyl-CoA:cholesterol acyltransferase 1 (ACAT1), yet a single-residue mutation sensitized ACAT1 for inhibition. Collectively, our studies provide a structural foundation for developing DGAT1 and other MBOAT inhibitors.

For many organisms, triacylglycerols (TGs) are a major storage form of metabolic energy as reduced carbon acyl chains, esterified to a glycerol backbone. Accumulation of TG in obesity and dysregulated TG metabolism are closely associated with metabolic disorders such as cardiovascular disease and diabetes[1–3]. Overproduction of TG is observed in various cancers[4–6], and evidence suggests a protective role of TG in promoting cancer cell survival and growth[7,8]. Therefore, inhibition of TG synthesis and storage represents a possible therapeutic strategy for TG-related diseases.

Acyl-CoA:diacylglycerol acyltransferase 1 (DGAT1) is one of two enzymes known to catalyze TG synthesis in humans[9]. It esterifies diacylglycerol (DAG) with acyl-CoA to generate TGs at the endoplasmic reticulum (ER) (Fig. 1a). In humans, this reaction is important for

metabolic energy storage, for instance, in the intestine, liver, and adipose tissue[10]. DGAT1 also protects the ER from the accumulation of bioactive lipids derived from excess fatty acids[11].

DGAT1 belongs to the family of membrane-bound O-acyltransferases (MBOAT), with members in all kingdoms of life[12]. Humans have 11 MBOATs that mediate the acylation of different lipids or proteins. Mammalian MBOATs (e.g., human DGAT1[13,14], ACAT1/2[15–17], HHAT[18,19], PORCN[20], MBOAT7[21], and chicken MBOAT5[22]) are saddle-shaped proteins with 9–12 transmembrane (TM) segments. Each enzyme has a catalytic center with evolutionarily conserved histidine and asparagine (aspartate for HHAT) residues that are embedded deeply within the ER membrane. A tunnel through the enzyme provides access to acyl-CoA to the catalytic site from the

[1]Department of Molecular Metabolism, Harvard T.H. Chan School of Public Health, Boston, MA, USA. [2]Department of Cell Biology, Harvard Medical School, Boston, MA, USA. [3]Department of Biochemistry and Biophysics, College of Agriculture and Life Sciences, Texas A&M University, College Station, TX, USA. [4]Department of Biochemistry and Molecular Biotechnology, University of Massachusetts Chan Medical School, Worcester, MA, USA. [5]Cryo-EM Core Facility, University of Massachusetts Chan Medical School, Worcester, MA, USA. [6]School of Life Sciences, Southern University of Science and Technology, Shenzhen, China. [7]Broad Institute of MIT and Harvard, Cambridge, MA, USA. [8]Cell Biology Program, Sloan Kettering Institute, Memorial Sloan Kettering Cancer Center, New York, NY, USA. [9]Howard Hughes Medical Institute, Boston, MA, USA. [10]These authors contributed equally: Maofu Liao, Robert V. Farese Jr., Tobias C. Walther. ✉e-mail: liaomf@sustech.edu.cn; robert@mskcc.org; twalther@mskcc.org

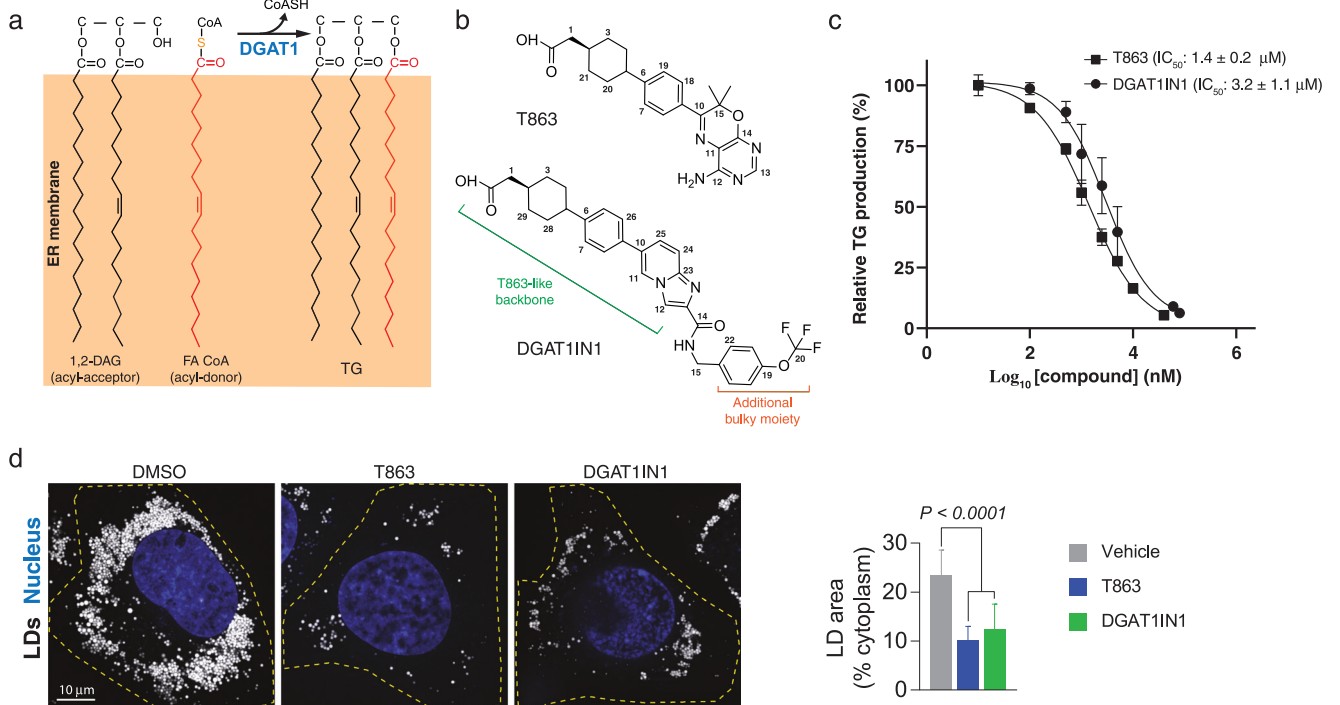

**Fig. 1 | Inhibition of TG biosynthesis by DGAT1 inhibitors. a** The DGAT1-mediated TG synthesis reaction. DGAT1 cleaves the thioester bond of fatty acyl-CoA (acyl-donor) and transfers the acyl chain to the hydroxyl group of diacylglycerol (acyl-acceptor) to form the TG product at the ER. **b** Chemical structures of two DGAT1 inhibitors, T863 and DGAT1IN1. **c** Dose-dependent inhibition curves of T863 and DGAT1IN1 for purified human DGAT1. The half maximal inhibitory concentration (IC$_{50}$) and standard deviation were calculated from three independent experiments ($n = 3$). The dose-response curve of each inhibitor shows results from one representative experiment; data points are shown as mean ± s.d., calculated from three technical replicates within this experiment. **d** T863 and DGAT1IN1 inhibit lipid droplet (LD) formation in vivo. T863 and DGAT1IN1 pre-treatment of SUM159 cells reduces LDs after incubating cells for 18 h in a 0.5 mM oleate-containing medium. Quantification of the LD area is shown as means ± sd, $n = 20$, 25, 20 cells from two independent experiments. See "Methods" for details. Statistical analysis by one-way ANOVA.

cytosol. In addition, large cavities open to the ER lumen or membrane to allow for access of protein or lipid acyl acceptor substrates to the catalytic site.

Numerous small molecule inhibitors of DGAT1 have been developed for treating metabolic diseases characterized by TG overaccumulation[23]. In clinical trials, some inhibitors were found to cause dose-related gastrointestinal side effects, limiting their utility for treating chronic metabolic disorders[24]. However, studies using these inhibitors in animal models also suggest they might provide therapeutics to treat cancers with altered TG metabolism, including glioblastoma and prostate cancer[7,25].

The molecular mechanisms of these compounds for inhibiting DGAT1 and TG synthesis remain uncertain. Moreover, our understanding of small-molecule inhibition of MBOAT enzymes, in general, is limited. Inhibitors have been developed for other MBOAT enzymes, such as PORCN[26] and HHAT[27], that acylate the signaling proteins Wnt and Shh, respectively, or MBOAT5[28] and MBOAT7[21] that acylate phospholipids. Some are in clinical trials[26,29]. But how much inhibitors achieve specificity for the different MBOAT enzymes and whether there are common themes for enzyme inhibition are not understood.

To address these questions, we investigated how potent and selective DGAT1 inhibitors block TG synthesis. Using cryo-EM, we resolved the complex structures of human DGAT1 with inhibitors T863 or DGAT1IN1, each at ~3.2 Å resolution. The structures, together with structure-based biochemical analyses, show how these compounds inhibit DGAT1 and provide insights into how they achieve MBOAT selectivity. The shared chemistry between the compounds and their binding modes further suggest chemical features to consider in developing MBOAT inhibitors.

## Results

### The DGAT1 inhibitors T863 and DGAT1N1 inhibit recombinant human DGAT1

T863 (Fig. 1b) was the first reported compound to inhibit DGAT1[30], and many subsequently developed inhibitors resemble its chemical structure (Supplementary Fig. 1a). The DGAT1 inhibitor DGAT1N1 has a similar scaffold but contains an additional bulky moiety connected to the T863 backbone[31] (Fig. 1b), suggesting a related mechanism of action (Supplementary Fig. 1b). Both T863 and DGAT1N1 were reported as potent and selective DGAT1 inhibitors in assays performed with cell lysates or extracted microsomes over-expressing human DGAT1[32]. We found that both compounds similarly inhibited purified human DGAT1 (Fig. 1c). Additionally, each inhibitor reduced lipid droplets (LDs) by ~50% in SUM159 cells incubated in a culture medium containing oleic acid (Fig. 1d).

### T863 inhibits DGAT1 by occupying the acyl-CoA binding site

To ascertain how T863 inhibits DGAT1, we determined the cryo-EM structure of DGAT1 bound to this compound when reconstituted in amphipol PMAL-C8. The resulting DGAT1 cryo-EM dimer map resolved overall at ~3.2 Å resolution and was very similar to the structure of apo DGAT1 (0.73 Å RMSD of 406 aligned residues)[13,14]. The structure contains an additional density corresponding to one T863 molecule bound to each DGAT1 protomer (Figs. 2a and 2d, Supplementary Figs. 2 and 3). T863 resided at the end of the cytosol-facing acyl-CoA-binding tunnel of DGAT1[33] (Fig. 2b). The positioning of T863 in acyl-CoA binding tunnel is consistent with previous results showing that the drug competes with the acyl-CoA substrate for enzyme binding (Fig. 2d)[32]. The EM density of T863 was mostly resolved, except for the carboxylate group (Fig. 2b, Supplementary Fig. 3d). The lack of density

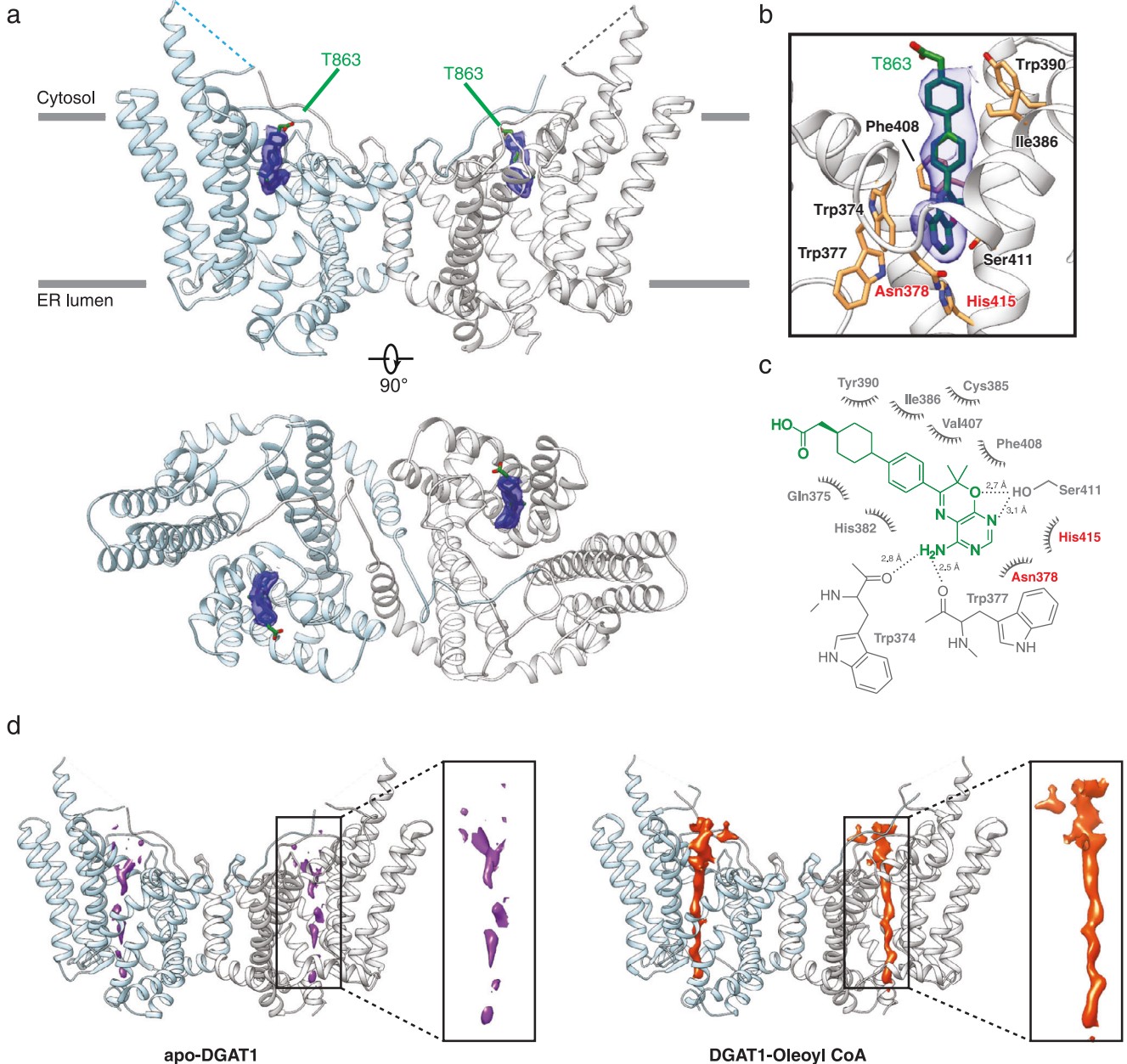

**Fig. 2 | Cryo-EM structure of human DGAT1 in complex with T863. a** Ribbon representation of the human DGAT1 structure bound to T863 (green sticks). The EM density of T863 is depicted as a blue surface. The dashed line indicates a disordered segment in DGAT1 that is not resolved in the cryo-EM map. The complex structure is shown along the membrane plane and from the cytosolic side. **b** Zoomed-in view of intermolecular interactions between T863 with DGAT1. The blue surface represents the EM density of T863. Residues interacting with T863 are shown in orange. The catalytic residues His[415] and Asn[378] are labeled in red. **c** Two-dimensional interaction diagram for T863. Polar interactions are shown with dashed lines. Residues participating in non-polar interactions with T863 are shown as spiked arcs. **d** Comparison of the EM density at the fatty acyl CoA binding tunnel in apo DGAT1 (purple surface in apo-DGAT1, EMDB: 21461, PDB: 6VYI), oleoyl CoA-bound DGAT1 (orange surface in DGAT1-Oleoyl CoA, EMDB: 21481, PDB: 6VZ1)[13]. The EM maps in each state are contoured at 3.5σ.

of this group may be due to its sensitivity to electron radiation damage during cryo-EM data collection[34].

T863 interacts with DGAT1 through both hydrophobic and polar interactions (Fig. 2b, c). Multiple residues critical for fatty acyl CoA binding[13] (e.g., Tyr[390], Ile[386], Val[407], Phe[408], Gln[375], and His[384]) form hydrophobic interactions with the cyclohexane and benzene rings of T863 (cyclohexylbenzene moiety) (Fig. 2c). For the bulky region of T863 that inserts into the fatty acid binding tunnel, the oxygen atom connecting C[14] and C[15] and the tertiary amine linking the C[13] and C[14] position form hydrogen bonds with the evolutionarily conserved residue Ser[411]. In addition, a primary amine at the C[12] position forms hydrogen bonds with the main chain carbonyl groups of Trp[374] and Trp[377] (Fig. 2c).

Furthermore, the bulky region (7,7-dimethyl-7H-pyrimido(4,5-b)[1,4] oxazin-4-amine) that links to the cyclohexylbenzene moiety interacts with the catalytic residues His[415] and Asn[378] through hydrophobic interactions (Fig. 2c). While the overall structure of DGAT1 appears unaltered by T863 binding, a few residues at the drug-binding pocket, including Phe[408] and Asn[465] had subtle conformation changes, orienting them toward the T863 molecule (Supplementary Fig. 4b).

**DGAT1N1 inhibits DGAT1 at the acyl-CoA binding pocket and interacts with the catalytic residues via an amide group**

The similar scaffold for T863 and DGAT1N1 suggests a similar mode of binding. However, this would require DGAT1N1's additional bulky

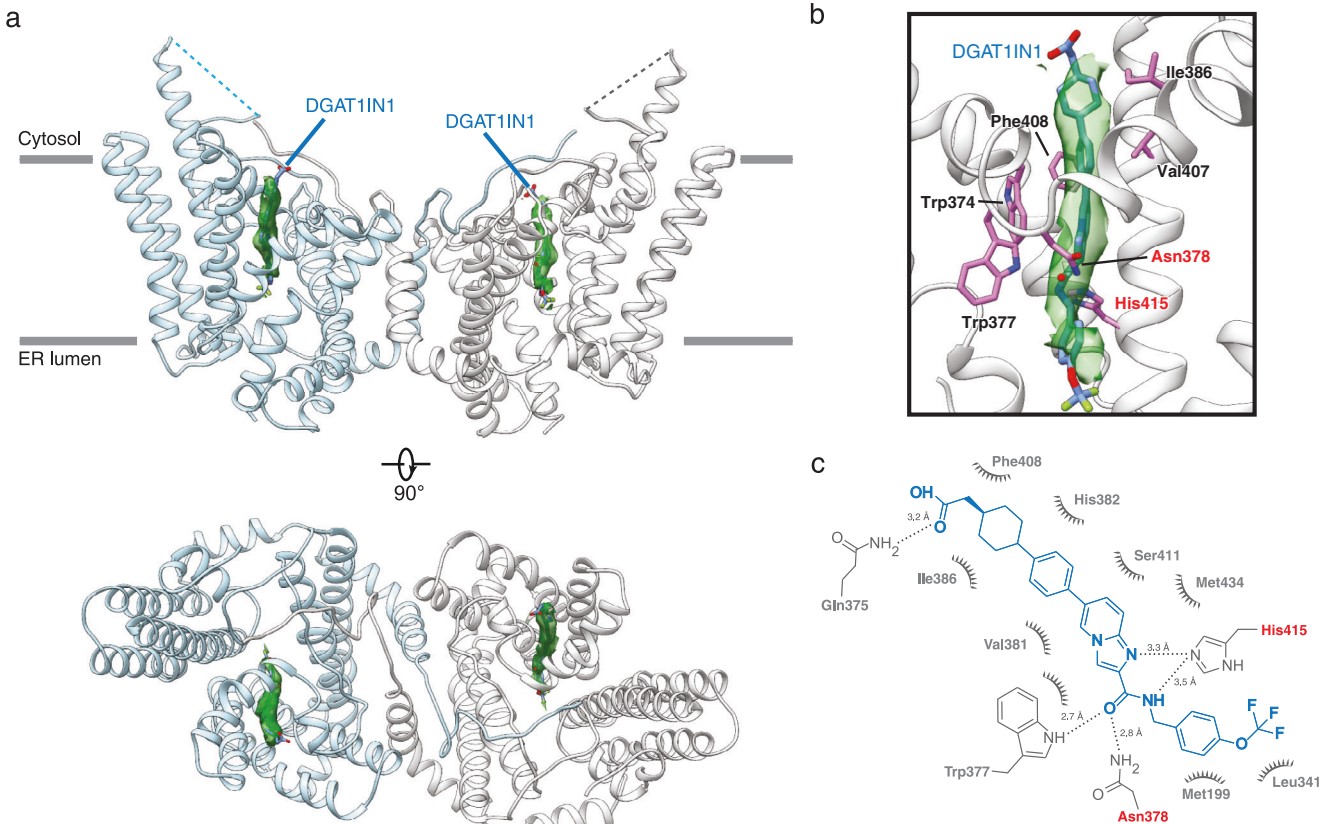

**Fig. 3 | Cryo-EM structure of human DGAT1 in complex with DGAT1IN1.**
**a** Ribbon representation of the human DGAT1 structure bound to DGAT1IN1 (blue sticks), viewed along the membrane plane and from the cytosolic side. The green surface represents the EM density of DGAT1IN1. **b** Zoomed-in view of intermolecular interactions between DGAT1IN1 with DGAT1. The EM density of DGAT1IN1 is shown as a green surface. Residues interacting with DGAT1IN1 are shown in purple. The catalytic residue His[415] and Asn[378] are labeled in red. **c** Two-dimensional interaction diagram for DGAT1IN1. Polar interactions are shown with dashed lines. Residues participating in non-polar interactions with the inhibitor are shown as spiked arcs.

(trifluoromethoxy)benzene moiety linked at the C[13] position to pass through the narrow fatty acyl CoA tunnel and enter the catalytic center (Fig. 1b). To determine if this is the case, we used cryo-EM to solve the structure of DGAT1 in complex with DGAT1IN1 with an overall resolution of ~3.2 Å (Supplementary Figs. 5 and 6). In this map, the DGAT1IN1 density is indeed positioned in the fatty acyl CoA binding tunnel (Fig. 3a) with most of the compound visible, except the carboxylate and the trifluoroethane ends of the molecule, likely due to a lack of interactions with DGAT1 or radiation damage[34] (Fig. 3b, Supplementary Fig. 6d). The portion of DGAT1IN1 that is the same as T863 is inserted ~1.2 Å deeper into the tunnel (Supplementary Fig. 7a, b), leading to different hydrophobic interactions with DGAT1 than observed for T863 (Fig. 3b,c).

By inserting deeper into the tunnel, DGAT1N1's (trifluoromethoxy) benzene moiety extends toward the catalytic center, buried deep within the enzyme. The amide group of DGAT1N1 connecting its T863 and bulky moieties between C[13] and C[15] forms extensive hydrogen bonds with the evolutionarily conserved Trp[377] and the catalytic His[415] and Asn[378] residues. This binding appears to lock the inhibitor into the reaction center (Fig. 3b, c). The inserted (trifluoromethoxy)benzene portion has only hydrophobic but no apparent polar contacts with DGAT1.

## Structural basis for selectivity of DGAT1 inhibitors

Human MBOATs share similar acyl-CoA binding sites, but inhibitors that may block these sites are known to be selective for different MBOATs[30,31,35]. To better understand how this is attained, we examined the specificity of inhibitors for DGAT1 and ACAT1, two enzymes

that catalyze the synthesis of neutral lipids. Among MBOAT enzymes, DGAT1 and ACAT1 share a high degree of similarity with ~3.8 Å RMSD for the conserved MBOAT-core region (TM2–TM9) within a protomer (Fig. 4a, b). Both enzymes catalyze analogous reactions, esterifying acyl-CoA with either diacylglycerol or cholesterol, respectively[9,36]. Yet, we found that inhibitors for these enzymes are highly selective– T863 did not inhibit the catalytic activity of ACAT1 in microsomes (Fig. 4c, Supplementary Fig. 8a). In contrast, the ACAT1 inhibitor ATR101 completely inhibited ACAT1 activity (Fig. 4d, Supplementary Fig. 8b).

We next investigated how T863 inhibits DGAT1 potently but has only minor inhibition of ACAT1. Comparing the structure of DGAT1 complexed with T863 with the ACAT1 structure showed a difference in the acyl-CoA binding tunnels of the two enzymes–TM8 in ACAT1 is tilted at the cytosolic opening compared with the position of the analogous helix of DGAT1 (Fig. 4a, b). In addition, a loop connecting TM8 with TM9 adopts a more extended structure in ACAT1 than in DGAT1 (Fig. 4b). Amino acids in this loop region are less evolutionarily conserved between ACAT1 and DGAT1 than in other regions of the proteins (e.g., TM8 and TM9) (Fig. 4e), and they form a more constricted entry site to the fatty acyl CoA binding tunnel in ACAT1 than in DGAT1 (Fig. 4a, b). Specifically, Asn[478] of TM8 in ACAT1 protrudes into the binding tunnel, where we hypothesized it generates a steric clash with the cyclohexylbenzene moiety of T863 (Fig. 4b, right panel) and interferes with inhibitor binding.

To test this hypothesis, we replaced ACAT1 Asn[487] with alanine, equivalent to the residue found in DGAT1 (Fig. 4b, e). Gel filtration and activity analyses indicated that the mutated protein was functional in

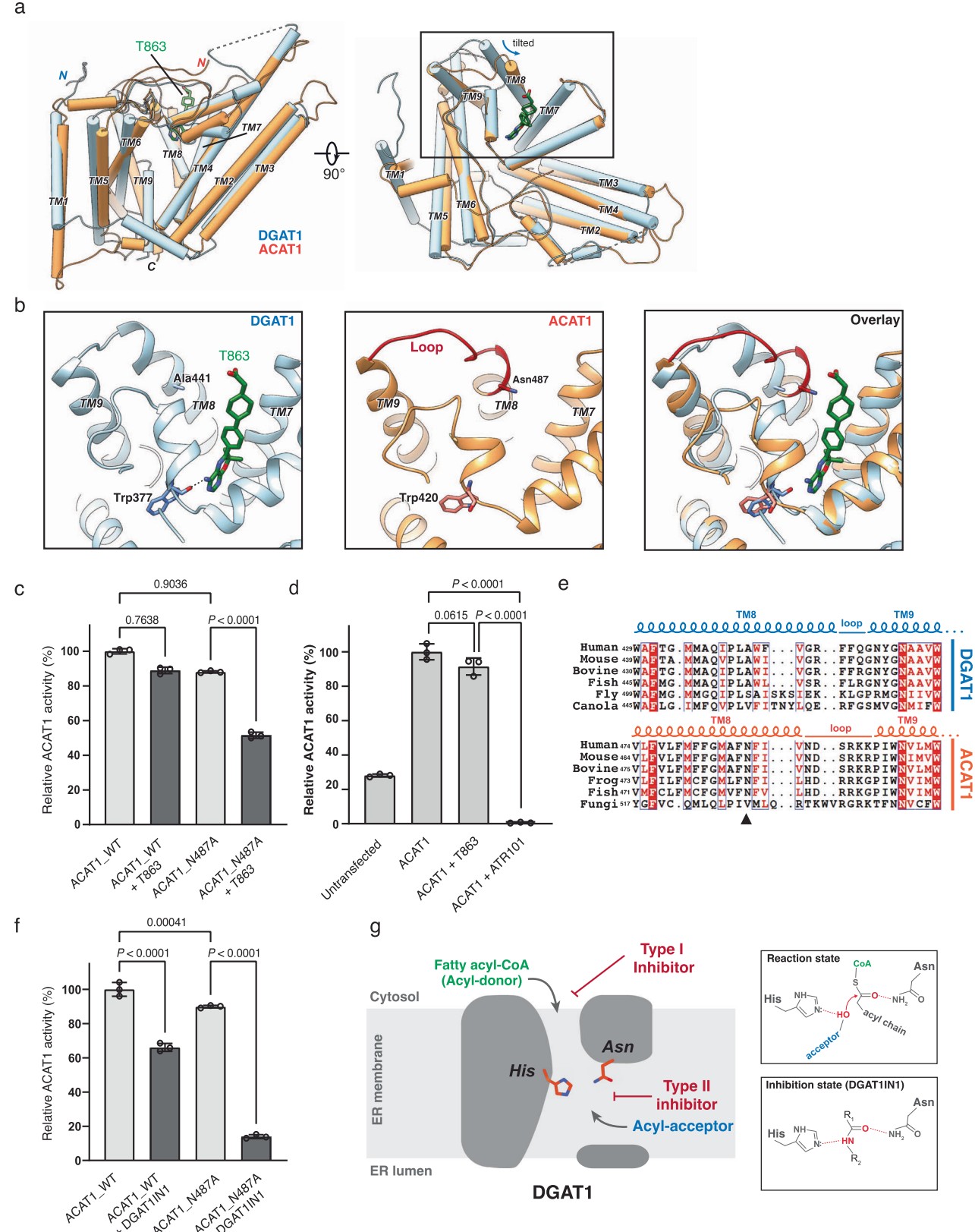

cholesterol esterification (Fig. 4f, Supplementary Fig. 8c). Yet, this single amino acid substitution rendered ACAT1 ~40% more sensitive to T863 inhibition (Fig. 4f, Supplementary Fig. 8c). Inhibition of ACAT1 N487A was less efficient than that for DGAT1. Possibly in DGAT1, T863 interacts with the main chain carbonyl of Trp[377], and the equivalent residue Trp[420] in ACAT1 may have a different orientation of the main

chain carbonyl group that weakens the interaction with T863 (Fig. 4b). To further test if residues of the loop region between TM8 and TM9 are crucial for drug binding, we replaced the Ala[441] in DGAT1 to Asn (which occupies the corresponding site in ACAT1) and tested the sensitivity of mutated DGAT1 to T863. Consistent with our hypothesis, replacing this non-conserved residue did not affect DGAT1 activity but

**Fig. 4 | Structural basis of inhibition selectivity of T863 and DGAT1IN1 for DGAT1 versus ACAT1. a** Structural superposition of monomeric DGAT1 (blue) and ACAT1 (orange, PDB: 6VUM)[16]. The structure of human DGAT1 and ACAT1 monomers are depicted as cylinders. TM transmembrane helix. The bound T863 in DGAT1 is also shown (green). **b** Detailed view of the T863 binding pocket in DGAT1 (left) and the corresponding region in ACAT1 (middle). Note the loop region labeled red in ACAT1 adopts a different conformation than that in DGAT1 (right). **c** T863 inefficiently inhibits ACAT1 but does so with higher potency for the Asn487Ala mutant. **d** ACAT1 inhibitor ART101 exhibits potent inhibition on ACAT1. **e** Sequence alignment of the red region in (**b**) among DGAT1 and ACAT1 proteins. The squiggles and solid lines on the top represent transmembrane helices and loop regions, respectively. The solid triangle denotes the Asn487 in ACAT1 protruding into the corresponding pocket in DGAT1 at T863 binding pocket. **f** DGAT1IN1 exhibits

inhibition on the wild-type ACAT1 and increased inhibition on the Asn487Ala mutant. Experiments in **c** and **f** were repeated three times independently ($n = 3$) with similar results. The data shown are one representative result. Data points are shown as mean ± s.d., calculated from three technical replicates. FFA free fatty acid, NS non-specific band. T863 and DGAT1IN1 were used at 10 μM, and ART101 was used at 1 μM in inhibition studies. Statistical analysis by one-way ANOVA (**c, d, f**). The raw TLC plates are shown in Supplementary Fig. 8a–c. **g** Proposed DGAT1 inhibition mechanism by small-molecule inhibitors. The catalytic His[415] and Asn[378] residues localize in the ER membrane. Type I inhibitors (e.g., T863 and DGAT1IN1) compete with the fatty acyl CoA substrate for binding at the cytoplasmic side of the enzyme; the type II inhibitors enter the catalytic center from the lateral opening. Amide bond-containing inhibitors may act by mechanism-based inhibition, locking the catalytic His and Asn residues (right panel).

significantly desensitized it for inhibition by T863 (Supplementary Fig. 8e).

DGAT1IN1 inhibited greater inhibition of ACAT1 than T863, and its inhibition was further enhanced for ACAT1 with the N487A mutation (Fig. 4f, Supplementary Fig. 8c). This higher potency of DGAT1IN1 towards ACAT1 is likely mediated by more efficient blocking of the active site with the amide bond and the (trifluoromethoxy)benzene bulky moiety.

## Discussion

Here, we report cryo-EM structures of human DGAT1 in complex with its inhibitors T863 or DGAT1IN1. Each of the compounds binds DGAT1 in the cytosolic part of the fatty acyl-CoA–binding tunnel and competes with substrate binding to block TG synthesis. Hydrophobic interactions in the cytosolic part of the substrate-binding tunnel and hydrogen bonds with evolutionarily conserved residues near the catalytic center mediate enzyme inhibition. Given that many DGAT1 inhibitors share structural features with T863 (Supplementary Fig. 1), this mode of inhibition is likely a common feature. DGAT1N1 extends an additional bulky end group more deeply into the catalytic center, where its amide group binds to the catalytic residues (His[415] and Asn[378]).

Inasmuch as all MBOAT enzymes appear to have conserved His and Asn residues in their catalytic core[12], the interaction of the amide group of DGAT1IN1 with this residue suggests it may be a more general pharmacophore for MBOAT enzymes. In support of this idea, interactions of the catalytic His and Asn with the amide bond are consistent with their proposed chemical roles during MBOAT-mediated reactions[15,16] (Fig. 4g). Additionally, a recent structure of PORCN bound to its inhibitor LGK974 at the acyl-CoA binding site suggests that the amide bond can also interact with non-catalytic residues of the reaction center to achieve potent inhibition[20]. A survey of compounds targeting DGAT1 or other MBOATs suggests that an amide bond in the middle region connected to two hydrophobic moieties ($R_1$ and $R_2$) is often a shared principle of these inhibitors (Supplementary Figs. 1b and 10). The $R_1$ and $R_2$ moieties adjacent to the amide moiety likely confer the selectivity for each MBOAT enzyme (Fig. 4g, Supplementary Fig. 10). Thus, these structural insights suggest this pharmacophore scaffold as a general strategy for the development of additional MBOAT inhibitors that block acyl-CoA binding.

Some DGAT1 inhibitors, such as XP620, appear to act via a different mechanism, possibly inhibiting DGAT1 at the membrane-embedded lateral gate that allows acyl-acceptor substrate binding (Supplementary Fig. 10)[37,38]. Similarly, ATR101 may inhibit ACAT1 at its cholesterol-binding pocket (Supplementary Fig. 9a–c)[16,39]. XP620 and ATR101 exhibit high selectivity on their targeted MBOAT enzymes (Supplementary Fig. 9d,e)[32]. Both drugs harbor bulkier and more lipophilic scaffolds and therefore are less likely to access the catalytic chamber via the narrow acyl-CoA binding tunnel (Supplementary Fig. 9f). Additionally, pyripyropene A, an ACAT2-specific inhibitor[40],

lacks an amide bond and shares some similarity to cholesterol, suggesting it may also access the catalytic site via the putative acyl-acceptor entrance at the lateral gate. Thus, the available evidence suggests that there are two classes of inhibitors for DGAT1, ACATs, and possibly other MBOATs: type I inhibitors compete for the acyl CoA binding from the cytosol (e.g., T863, DGAT1IN1, LGK974), and type II inhibitors enter the catalytic center from the lateral gate open to the membrane leaflet to block acyl acceptor substrates binding (e.g., XP620, ATR101, and possibly pyripyropene A) (Fig. 4g, Supplementary Fig. 9f).

Finally, our studies provide insights into the mystery of how MBOAT inhibitors achieve target selectivity. We find that although DGAT1 and ACAT1 closely resemble each other and have a common acyl-CoA binding tunnel[13–17], the tested DGAT1 inhibitors showed little activity against ACAT1 (Fig. 4c, d, f), apparently due to a steric clash with a loop of ACAT1 that blocks the entrance of these type I DGAT1 inhibitors to the acyl-CoA binding tunnel. However, a single-point mutant of ACAT1 was sufficient to enable the DGAT1 inhibitors to be active against ACAT1 (Fig. 4c, d, f). Conversely, mutating a single residue in this region of DGAT1 (A441N) was sufficient to reduce the potency of T863 for DGAT1 inhibition (Supplementary Fig. 8e). This suggests that although all MBOATs bind to acyl-CoA substrates, the binding pocket of each MBOAT is sufficiently different to enable inhibitor selectivity. Conversely, it also suggests that cross-reactivity of inhibitors for different MBOATs may exist in some cases and should be considered in future MBOAT inhibitor development.

## Methods
### DGAT1 expression and purification

The expression and purification of DGAT1 were conducted according to published protocols[13]. Briefly, human DGAT1 (Uniport ID: O75907) fused with an N-terminal maltose-binding protein (MBP), and a TEV cleavage site was expressed in HEK293 GnTi- (ATCC, CRL-3022) cells using the BacMam system. The bacmid was produced in DH10Bac *E. coli* cells (Thermo Fisher Scientific, 10361012). Baculovirus was generated by transfecting *Spodoptera frugiperda* (Sf9) cells (Expression Systems, 94-001S) and amplified by transfecting Sf9 cells for two rounds for large-scale transfection. To express MBP-DGAT1, HEK293 GnTi- cells were grown in suspension at 37 °C in FreeStyle 293 Expression Medium (ThermoFisher) with 1% FBS and 1× Glutamax solution (Gibco). When the cell density reached $3 \times 10^6$ cells per ml, baculovirus was added to the culture at 5% (v/v) final concentration. After 15 h of infection at 37 °C, the culture was supplemented with 10 mM sodium butyrate to boost the expression. After further growth for ~ 36 h at 25 °C, cells were harvested and washed in PBS buffer. Cell pellets were flash-frozen in liquid nitrogen and stored at −80 °C or placed on ice for immediate use. DGAT1 Asn487Ala mutant was generated by site-directed mutagenesis (forward primer 5′ to 3′: gatggct cagatcccactgaactggttcgtgggc, reverse primer 5′ to 3′: gaaaaagcggcc cacgaaccagttcagtgggatctg), following similar procedures for ACAT1 (see below). The DGAT1 wild type and mutant enzymes were purified

by using the 8×His tag at N-termini by following the purification scheme described below.

All protein purification steps were performed at 4 °C. Cell pellets collected from 1 L of cell culture were resuspended in 80 mL of TSGE buffer (50 mM Tris-HCl, pH 8.0, 400 mM NaCl, 10% v/v glycerol and 1 mM EDTA), supplemented with protease inhibitor cocktail (Roche). DGAT1 inhibitors T863 or DGAT1IN1 (MedChemExpress) were added at 5 μM during all purification steps. To lyse the cells, GDN (Anatrace) was added into the solution with a final 0.5% w/v concentration, and the mixture was generally agitated for 1 h before MgCl₂ was added to the final concentration of 5 mM. The mixture was then agitated for 1 h more. The insoluble debris was removed by centrifugation, and the supernatant was incubated with prewashed amylose resin for 2 h. The resin was washed to remove contaminant proteins. The MBP-DGAT1 was finally eluted in TSGE buffer with 20 mM maltose and 0.1% w/v digitonin (Sigma-Aldrich). The eluate was concentrated, and the MBP tag was cleaved off by TEV protease digestion. The protein was then further purified by gel filtration chromatography in Superose 6 column (GE Healthcare) in TSM buffer (50 mM Tris-HCl, pH 7.5, 400 mM NaCl, 10 mM MgCl₂) supplemented with 0.05% w/v digitonin. The peak fractions containing DGAT1 were collected and concentrated. To prepare cryo-EM samples, the purified DGAT1 in detergent was mixed with PMAL-C8 (Anatrace) at a 1:5 (w/w) ratio in the presence of a 5 μM inhibitor followed by gentle agitation overnight in a cold room. The detergent was then removed with Bio-Beads SM-2 (Bio-Rad). The solution was cleared by passing a 0.22-μm filter before the final gel filtration purification in TSM buffer. The peak containing DGAT1 was collected for cryo-EM studies.

### ACAT1 expression and microsome extraction

The human ACAT1 gene (Uniport ID: P35610) was cloned into a pFastBacMam vector constructed with the mammalian CMV promoter and an N-terminal GFP protein. The ACAT1 mutants were generated by the QuickChange Site-Directed Mutagenesis kit (Agilent) per the manufacturer's instruction (forward primer 5′ to 3′: ctcttcatgttctttggaatggctttcgccttcattgtc, reverse primer 5′ to 3′: ctttttccgactatcattgacaatgaaggcgaaagc). To express wild-type or mutated ACAT1, 0.5 mg pFastBacMam plasmids harboring the intended construct were incubated with 1.5 mg polyethylenimines (Polysciences) in 40 mL FreeStyle 293 Medium (ThermoFisher) for 30 min at room temperature. After incubation, the mixture was added into 250 mL HEK293F cell suspension at a density of $3 \times 10^6$ cells per ml. Transfected cells were grown for 48 h at 37 °C before harvest.

To extract ACAT1 overexpression microsomes, cell pellets from 250 mL transfected culture were harvested, washed with PBS, and resuspended in 30 mL homogenizer buffer (250 mM sucrose, 20 mM Tris-HCl pH7.4, 1 mM EDTA) with protease inhibitor cocktail (Roche). The resuspended cells were transferred to a glass dounce homogenizer prechilled on ice. Cells were broken manually by stroking a tight-fitting pestle by 40–50 times, and unbroken cells were removed by centrifugation at 600g for 5 min. The remaining cell debris in the supernatant was further clarified by centrifugation at 8000g for 10 min. Finally, the microsomes were collected by centrifugation of the supernatant at 100,000g for 1 h. All centrifugation steps were performed at 4 °C. The microsome pellets were resuspended in 4 mL of ice-cold homogenizer buffer containing 10% glycerol (v/v). Microsome concentration was determined by measuring the absorbance at OD280 nm. Then the microsomes were frozen in liquid nitrogen and stored at −80 °C for further use. To analyze the expression level of wild-type and mutated ACAT1, the microsomes were solubilized in solubilization buffer (20 mM Tris-HCl pH8.0, 150 mM NaCl) supplemented with 1% DDM (w/v) in a cold room for 1 h. The solubilized mixture was filtered into a 0.22 μm filter and analyzed by fluorescence-detection size-exclusion chromatography (FSEC) by monitoring the fluorescent signal from the GFP tag fused with the ACAT1 proteins.

### DGAT1 and ACAT1 activity assays

The activity of DGAT1 was determined using published methods[13]. The half maximal inhibitory concentration (IC₅₀) of T863 or DGAT1IN1 inhibitor was determined using purified DGAT1 enzymes in the digitonin-containing buffer. For ACAT1, the activity was determined by using ACAT1 overexpressing microsomes. Briefly, the reaction was performed in a 200 μL reaction mixture containing 300 mg of ACAT1 microsomes, 20 mM Tris-HCl, pH 8.0, 200 mM KCl, 150 mM NaCl, 0.25 g/L of delipidated BSA, 0.005% (w/v) GDN, 0.25% (w/v) CHAPS and 50 μM oleoyl-CoA containing 0.2 μCi [¹⁴C]-oleoyl-CoA as a tracer (American Radiolabeled Chemicals). The reaction mixture was incubated on a bench-top thermomixer at 500 rpm, 25 °C for 30 min in the presence of an inhibitor, as denoted in the main text. To initiate the reaction, cholesterol substrate prepared in phosphatidylcholine micelle[41] was added into the reaction mixture to a final concentration of 100 μM. Reactions were carried out at 37 °C with gentle agitation for 30 min or as indicated. The reactions were quenched by adding 500 μL of chloroform/methanol (2:1 v:v) followed by 2% phosphoric acid for phase separation. The organic phase was collected, dried, resuspended in chloroform, and loaded on a silica gel TLC plate (Analtech). Lipids were resolved in a solvent system consisting of hexane, diethyl ether, and acetic acid (80:20:1 v:v:v). The radioactivity of oleoyl cholesterol bands was revealed by Typhoon FLA 7000 phospho-imager (GE Healthcare Life Sciences) and quantified by Quantity One (V4.6.6). The activity of mutated ACAT1 was analyzed in the same manner. The final activities of wild-type ACAT1 or the mutants were normalized to their expression level in the microsome determined by FSEC analyses as described in the above section. One-way ANOVA with Dunnett's post hoc test was applied for analyses of radioactive products generated by DGAT1 and ACAT1.

### Fluorescence microscopy

SUM159 breast cancer cells were obtained from Dr. Tomas Kirchhausen (Harvard Medical School) and grown in DMEM/F-12 GlutaMAX (Life Technologies) with 5 μg/ml insulin (Cell Applications), 1 μg/mL of hydrocortisone (Sigma), 5% FBS (Life Technologies, Thermo Fisher), 50 μg/mL of streptomycin, and 50 U/mL of penicillin. For LD area experiments, cells were treated with 10 μM T863 or DGAT1IN1 (or equivalent volume of DMSO) 30 min prior to the incubation in 0.5 mM oleic acid (complexed with essentially fatty acid-free BSA) containing medium. To determine LD formation, Cells were grown to 60–70% confluency on a 35-mm dish with a 14-mm No. 1.5 coverslip bottom (MatTek Life Sciences, #P35G-1.5-14-C). After appropriate inhibitor and oleic acid treatment, LDs were stained with BODIPY 493/503 (D3922, Thermo Fisher) and nucleus with Hoechst 33342 (H3570, Thermo Fisher).

For spinning disk confocal microscopy, a Nikon Eclipse Ti inverted microscope was used that features a CSU-X1 spinning disk confocal (Yokogama) and Zyla 4.2 PLUS scientific complementary metal-oxide semiconductor (sCMOS) (Andor, UK) was used. NIS-elements software (Nikon) was used for acquisition control. Plan Apochromat VC 100× oil objective (Nikon) with 1.40 NA was used, resulting in a 0.065-μm pixel size. Solid state excitation lasers−405 nm (blue; Andor), 488 nm (green; Andor), 560 nm (red; Cobolt), and 637 nm (far-red; Coherent)− shared quad-pass dichroic beam splitter (Di01-T405/488/568/647, Semrock), whereas emission filters were FF01-452/45, FF03-525/50, FF01-607/36, and FF02-685/40 (Semrock), respectively.

### Quantification of fluorescence images

Confocal images were quantified using ImageJ[42]. A series of 7 or 8 images were obtained randomly with a 100x objective for each condition, and cells that were visualized entirely within the field were selected for analysis, resulting in a sampling of 8 and 12 cells for a vehicle, 11 and 14 cells for T863, and 11 and 9 cells for DGAT1IN1, respectively, from each experiment. A cell boundary was drawn based

on the residual BODIPY staining, and within the cell boundary, an auto-thresholding method (Otsu) was applied to create an LD mask. A cytoplasm mask was created by subtracting the nucleus (using the auto-thresholding Huang method to the Hoechst channel) from the cell boundary. Finally, the LD area (% cytoplasm) was calculated by dividing the area of the LD mask by the area of the cytoplasm mask.

### EM sample preparation and data acquisition

To prepare the cryo-EM sample for structure determination, 2–3 μL of purified DGAT1 in PMAL-C8 with 5 μM inhibitor was applied to Quantifoil holey carbon grid (Cu R1.2/1.3; 400 mesh) glow discharged for 30 s. Optimal particle distribution was obtained with a protein concentration of 4–5 mg/mL. The grids were blotted with a Whatman #1 filter paper for 5 s with ~95% humidity at 4 °C and plunged frozen in liquid ethane using an FEI Vitrobot Mark IV system (Thermo Fisher Scientific). Cryo-EM data were collected on a Talos Arctica or a Titan Krios electron microscope (Thermo Fisher Scientific). Images were recorded using SerialEM[43]. Refer to Supplementary Table 1 for detailed information about microscope type and data collection parameters.

### EM data processing

Drift corrections were performed using MitionCor2[44], and images were binned 2 × 2 by Fourier cropping. The defocus values were determined using CTFFIND4[45] and motion-corrected sums without dose-weighting. Motion-corrected sums with dose-weighting were used for all other image processing. Particle picking was performed using a semi-automated procedure[46]. The 2D classification of selected particle images was performed by samclasscas.py, which uses SPIDER operations to run 10 cycles of correspondence analysis, $K$-means classification, and multireference alignment, or RELION 2D classification[47,48]. Initial 3D models were generated by using published DGAT1 cryo-EM maps[13]. 3D classification and refinement were performed using relion3_refine_mpi in RELION. All refinements followed the gold-standard procedure in which two-half of datasets were refined independently. The overall resolutions were estimated based on the gold-standard Fourier shell correlation (FSC) = 0.143 criterion. Local resolution variation of cryo-EM maps was calculated using ResMap[49]. The amplitude information of the final maps was corrected by applying a negative B factor using relion_postprocessing with the –auto_bfac option. The number of particles in each dataset and details related to data processing are summarized in Supplementary Table 1 and Supplementary Figs. 2 and 5.

### Model building and refinement

The DGAT1 density maps in MRC/CCP4 format were converted to the structure factors MTZ format in PHENIX[50]. The published DGAT1 dimer structure (PDB: 6VYI) was used as the initial model, and real-space refinement was performed by using the cryo-EM map[13]. The refined model was visually inspected and adjusted in COOT[51], and the resulting model was put back through the real-space refinement procedure for further refinement until the model reached an optimal geometry. Visual inspection of the cryo-EM map and the refined model clearly showed the EM density for the inhibitors. Atomic coordinates and geometric restraints for the T863 and DGAT1IN1 inhibitors were generated using the GRADE Web Server (Global Phasing) and manually fitted into the density in COOT. Then the DGAT1-inhibitor composite model was further refined in an iterative manner until it reached optimal geometric statistics as evaluated by MolProbity[52]. The DGAT1-inhibitor interactions were analyzed by LigPlot[53]. All cryo-EM figures were prepared in UCSF Chimera[54].

### Reporting summary

Further information on research design is available in the Nature Portfolio Reporting Summary linked to this article.

## Data availability

The 3D cryo-EM density maps generated in this study have been deposited in the Electron Microscopy Data Bank (EMDB) under accession numbers EMD-28577 (DGAT1-T863 complex) and EMD-28594 (DGAT1-DGAT1IN1 complex). The coordinates have been deposited into the Protein Data Bank (PDB) with accession numbers 8ESM (T863 bound) and 8ETM (DGAT1IN1 bound). This study also cited the published structure of human DGAT1 6VYI and EMD-21461, the oleoyl CoA bound state 6VZ1 and EMD-21481, and the human ACAT1 structure 6VUM and EMD-21390. Source data are provided in this paper.

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

## Acknowledgements

We thank members of the Liao and Farese & Walther laboratories for helpful discussions and Gary Howard for editorial assistance. This work was supported by National Institutes of Health grant R01 GM124348 (to R.V.F.) and the Howard Hughes Medical Institute, where T.C.W. is an investigator.

## Author contributions

X.S., M.L., T.C.W., and R.V.F. conceived the project. X.S. performed protein expression, purification, and reconstitution into amphipol, prepared cryo-EM grids, processed cryo-EM data, and built the atomic models. K.S. and X.C. screened cryo-EM grids and collected the data. X.S. performed the activity studies, K.W. and C.W.L. assisted with protein expression and purification, radioactive assay, and quantification. J.S. performed lipid droplet formation analysis. X.S., R.V.F., and T.C.W. wrote the paper. All authors analyzed and discussed the results and contributed to the paper.

## Competing interests

All authors declare no competing interests.
