## [Peer Review File · Nature Communications]

Mechanism of action for small-molecule inhibitors of triacylglycerol synthesisREVIEWER COMMENTS

Reviewer #1 (Remarks to the Author):

Membrane bound O-acyltransferase (MBOAT) is a class of membrane enzymes which catalyze the transfer of fatty acid chains to either lipid or protein substrate. DGAT1, an MBOAT, catalyzes the synthesis of triacylglycerols (TGs) which serve as the major energy storage form in adipose tissue. Although the structures of DGAT1 were reported in 2020, the regulatory and inhibitory mechanism of this important enzyme remains unclear. In this manuscript, Su et al. report the cryo-EM structures of DGAT1 with two different inhibitors, T863, and DGAT1IN1. The structures reveal that both inhibitors block the access of the acyl-CoA substrate. Interestingly, structural comparison with the previous findings on acyl-CoA: cholesterol acyltransferase 1 (ACAT1), along with their functional analysis, demonstrate that an ACAT1 mutant, N478A, presents a stronger sensitivity for the inhibitor DGAT1IN1 than the wild-type ACAT1 revealing the mechanistic basis of the specificity of DGAT1 inhibitors.

The cryo-EM maps and models look reasonable and support the conclusions as well. The only flaw is that neither cryo-EM map could completely cover the ligand. It may be due to the low local resolution, the sensitivity to electron radiation damage during data collection, or a lack of interactions between the related group and residues of DGAT1. Overall, the entire story is very appealing and will be in the interest of the general readership of Nature Communications. As the manuscript is well written, and the scientific quality deserves publication in Nature Communications, I suggest publishing this work if the authors address the following concerns or suggestions.

-Fig. 1c, could authors indicate the IC50 in the figure directly?

-PAGE 6, The last sentence of the penultimate paragraph, "binding (e.g., XP620, ATR101) (Fig. 4d, Extended Data Fig. 9f)." Fig.4d should be Fig. 4h.

-PAGE 6, "A survey of other compounds targeting DGAT1 or other MBOATs suggests that an amide bond in the middle region connected to two hydrophobic moieties is a shared principle of these inhibitors." This point is true, but the authors may also mention "PPPA (Ohshiro et al., *Arterioscler. Thromb. Vasc. Biol.* 2011)", a specific inhibitor of ACAT2 but without the amide bond.

-Fig. 4a, the color code should be indicated.

-Many Porcn inhibitors also disrupt the acyl-CoA binding to abolish the reaction (Yu, J. et al., *J. Cell Sci.* 2021 and Liu et al., *Nature* 2022). A structural comparison of inhibitor-bound Porcn and DGAT1 structures may highlight a general principle of type-I inhibitors in MBOAT inhibition.

Reviewer #2 (Remarks to the Author):

The manuscript by Sui et al. extends their previous cryoEM structural determination of DGAT1 by reporting the binding sites of two selective DGAT1 inhibitors, providing new insights into the mechanism of pharmacological inhibition. The experiments are straight forward and the conclusions are well supported by the data.

Minor concerns regarding imaging data and statistics:

Fig 1 states data are three replicates, but does not mention experimental reproducibility. X axis is odd (log 10 nM), suggest more standard -LogM.

Figure 1D. At 50% confluency there are thousands of cells per coverslip. While there is no reason to doubt the result, why were only 20-30 cells evaluated? How were these cells distributed between

independent experiments? Was cell selection performed in a blinded fashion?

Several legends variously state that the data are Mean \pm s.d., n = 3 independent experiments, noting that experiments were repeated three times with similar results. It is important to clearly/explicitly state what is being represented. It appears that figures are technical replicates of a single experiment that was performed three times with similar results. Technical replicates only state the reproducibility within a single experiment, not the general reproducibility of the phenomenon. Were statistics performed on technical replicates? If not, how were independent experiments normalized to 100%? n should refer to the statistical analysis of independent experiments, not technical replicates.

Is figure S8a a composite?

Regarding presentation:

The results of ligand inhibition require an understanding of how acyl-CoA binds. The presentation would be improved for the interested reader (but not expert) by incorporating elements of figure S4a into F2. Are there experimental data showing the compound inhibition is competitive with acyl-CoA? If so, please cite.

What does the N487A mutation do to the structure as predicted by MDS? Does it simply block the channel entrance or produce effects on the ligand binding pocket. Some backup by docking simulations might be helpful. Also, the ms would be strengthened by loss of function mutations in DGAT1 (e.g., Asn378 for DGATIN1) given the proposed importance on binding and catalysis.

Pharmacologist readers might have a problem with the liberal use of "specific", when "selective" is really meant (given that DGAT2IN1 clearly exhibits partial inhibition of ACAT1 at 10 μ M. Similarly, potency claims are usually backed up by dose-response data that determine potency (IC50) and efficacy (maximal inhibition).

REVIEWER COMMENTS

Reviewer #1 (Remarks to the Author):

Membrane bound O-acyltransferase (MBOAT) is a class of membrane enzymes which catalyze the transfer of fatty acid chains to either lipid or protein substrate. DGAT1, an MBOAT, catalyzes the synthesis of triacylglycerols (TGs) which serve as the major energy storage form in adipose tissue. Although the structures of DGAT1 were reported in 2020, the regulatory and inhibitory mechanism of this important enzyme remains unclear. In this manuscript, Sui et al. report the cryo-EM structures of DGAT1 with two different inhibitors, T863, and DGAT1IN1. The structures reveal that both inhibitors block the access of the acyl-CoA substrate. Interestingly, structural comparison with the previous findings on acyl-CoA: cholesterol acyltransferase 1 (ACAT1), along with their functional analysis, demonstrate that an ACAT1 mutant, N478A, presents a stronger sensitivity for the inhibitor DGAT1IN1 than the wild-type ACAT1 revealing the mechanistic basis of the specificity of DGAT1 inhibitors.

The cryo-EM maps and models look reasonable and support the conclusions as well. The only flaw is that neither cryo-EM map could completely cover the ligand. It may be due to the low local resolution, the sensitivity to electron radiation damage during data collection, or a lack of interactions between the related group and residues of DGAT1. Overall, the entire story is very appealing and will be in the interest of the general readership of Nature Communications. As the manuscript is well written, and the scientific quality deserves publication in Nature Communications, I suggest publishing this work if the authors address the following concerns or suggestions.

We thank this reviewer for their critical and constructive feedback on our manuscript. In response to the reviewer's critique, our manuscript has undergone revisions requested by the reviewers and the editor.

-Fig. 1c, could authors indicate the IC50 in the figure directly?

The IC50 for each inhibitor is now labeled in Fig. 1c. We also updated the standard deviations shown in the figure calculated from three independent measurements.

-PAGE 6, The last sentence of the penultimate paragraph, "binding (e.g., XP620, ATR101) (Fig. 4d, Extended Data Fig. 9f)." Fig.4d should be Fig. 4h.

We corrected this in the manuscript.

-PAGE 6, "A survey of other compounds targeting DGAT1 or other MBOATs suggests that an amide bond in the middle region connected to two hydrophobic moieties is a shared principle of these inhibitors." This point is true, but the authors may also mention "PPPA (Ohshiro et al., Arterioscler. Thromb. Vasc. Biol. 2011)", a specific inhibitor of ACAT2 but without the amide bond.

Thank you for this insightful comment. We have edited the manuscript to include this point in the Discussion (line #221-223, #227-229).

-Fig. 4a, the color code should be indicated.

We have added the color code to Fig. 4a.

-Many Porcn inhibitors also disrupt the acyl-CoA binding to abolish the reaction (Yu, J. et al., J. Cell Sci. 2021 and Liu et al., Nature 2022). A structural comparison of inhibitor-bound Porcn and DGAT1 structures may highlight a general principle of type-I inhibitors in MBOAT inhibition.

We thank the reviewer for this interesting point. Indeed, the inhibitor LGK974 appears to bind the acyl-CoA binding site of PORCN, and intriguingly it also contains an amide bond, which likely interacts with several residues surrounding the active site. We have edited the Discussion to include this point (line #207-209, #226-227).

Reviewer #2 (Remarks to the Author):

The manuscript by Sui et al. extends their previous cryoEM structural determination of DGAT1 by reporting the binding sites of two selective DGAT1 inhibitors, providing new insights into the mechanism of pharmacological inhibition. The experiments are straightforward, and the conclusions are well supported by the data.

We thank this reviewer for their critical and helpful evaluation of our manuscript. We have revised our manuscript in the response to the reviewer's critique.

Minor concerns regarding imaging data and statistics:

Fig 1 states data are three replicates, but does not mention experimental reproducibility. X axis is odd (log 10 nM), suggest more standard -LogM.

We corrected this in the updated Fig. 1c.

Figure 1D. At 50% confluency there are thousands of cells per coverslip. While there is no reason to doubt the result, why were only 20-30 cells evaluated? How were these cells distributed between independent experiments? Was cell selection performed in a blinded fashion?

A series of 7 (experiment #1) or 8 (experiment #2) images were obtained randomly with a 100x objective for each condition, and cells that were visualized entirely within the field were selected for analysis, resulting in the sampling of 8 and 12 cells for vehicle, 11 and 14 cells for T863, and 11 and 9 cells for DGAT1IN1 respectively from each experiment. Based on our previous studies, this method of analysis gives reproducible LD quantification. This detail has now been added to the manuscript for clarification (line #528-537). Our lab has performed multiple similar analyses using an automated confocal microscope that captures hundreds of cells at once in an unbiased manner, but this alternative method is limited due to poorer resolution inherent to the machine, variability in plate depth, and fixation step that often distorts LD morphology and staining. Although done manually, we found that the method used in this current experiment to be a more accurate and precise measurement of LD area, and given the obvious difference in the conditions tested, a relatively small number of cells was required to achieve statistical power.

Several legends variously state that the data are Mean \pm s.d., n = 3 independent experiments, noting that experiments were repeated three times with similar results. It is important to clearly/explicitly state what is being represented. It appears that figures are technical replicates of a single experiment that was performed three times with similar results. Technical replicates only state the reproducibility within a single experiment, not the general reproducibility of the phenomenon. Were statistics performed on technical replicates? If not,

how were independent experiments normalized to 100%? n should refer to the statistical analysis of independent experiments, not technical replicates.

We thank the reviewer for pointing this out.

To calculate the IC₅₀ of each drug, we performed dose-response inhibition assay three times. The IC₅₀ and its standard deviation (SD) for each compound were calculated from three dose-response curves performed independently (n=3). The IC₅₀ and SD are now labeled in Fig. 1b. The dose-response curves shown Fig.1b are one representative result from three independent experiments. For each data point in the curve, we measured the activity in triplicate (technical replicates) to calculate the mean and SD.

We have updated the manuscript to clarify this point (line #257-262).

Is figure S8a a composite?

No, this figure is an autoradiograph of a thin layer chromatography plate with 12 adjacent lanes. Lipids bands were assigned based on co-migration with unlabeled lipid standards. We have now clarified this in the legend (line #383-384). Please also note that Fig. S8a is now moved to Fig. S8b.

Regarding presentation:

The results of ligand inhibition require an understanding of how acyl-CoA binds. The presentation would be improved for the interested reader (but not expert) by incorporating elements of figure S4a into F2. Are there experimental data showing the compound inhibition is competitive with acyl-CoA? If so, please cite.

We thank the reviewer for this suggestion. We have incorporated previous Figure S4a into Fig.2d. In addition, we have clarified the manuscript to better explain previous biochemical studies on T863 and DGAT1 that suggested the drug competes with acyl-CoA but not diacylglycerol, which is consistent with our structural studies (line #113-115). We also cited this paper (PMID: 21990351) in our manuscript.

What does the N487A mutation do the structure as predicted by MDS? Does it simply block the channel entrance or produce effects on the ligand binding pocket. Some backup by docking simulations might be helpful. Also, the ms would be strengthened by loss of function mutations in DGAT1 (e.g., Asn378 for DGATIN1) given the proposed importance on binding and catalysis.

We thank the reviewer for this suggestion. Based on our structural and biochemical data, we hypothesize that the N487A mutant primarily relieves the steric hindrance for the compound to access the acyl-CoA binding pocket in ACAT1. To further test this, we followed the reviewer's suggestion by introducing an Asn residue in ACAT1 into DGAT1 at the corresponding position (Ala441). Based on our hypothesis, we expected the DGAT1_A441N mutant to exhibit a decreased inhibition by T863 because the bulkier Asn would sterically interfere with drug binding. Indeed, the DGAT1_A441N showed significant resistance to the T863 at all three tested T863 concentrations (1uM, 10uM, and 100 uM). Thus, we think steric hindrance is the most likely explanation (rather than stability in the pocket). These data have been added to the manuscript (line #182-186, and Extended Data Fig. 8e).

Pharmacologist readers might have a problem with the liberal use of "specific", when "selective" is really meant (given that DGAT2IN1 clearly exhibits partial inhibition of ACAT1 at 10 uM. Similarly, potency claims are usually backed up by dose-response data that determine potency (IC₅₀) and efficacy (maximal inhibition).

Thank you for this insightful comment. We have edited the manuscript in multiple places to indicate “selectivity” (instead of “specificity”), where appropriate. We agree with the referee on the use of “potency” and have edited the manuscript accordingly.

REVIEWERS' COMMENTS

Reviewer #2 (Remarks to the Author):

The authors have addressed all concerns.